# FOXO3 Deficiency in Neutrophils Drives Colonic Inflammation and Tumorigenesis

**DOI:** 10.3390/ijms24119730

**Published:** 2023-06-04

**Authors:** Jenisha Ghimire, Rida Iftikhar, Harrison M. Penrose, Patricia Snarski, Emmanuelle Ruiz, Suzana D. Savkovic

**Affiliations:** Department of Pathology and Laboratory Medicine, Tulane University School of Medicine, New Orleans, LA 70112, USA

**Keywords:** IBD, colon cancer, PMNs, LDs, FOXO3

## Abstract

Inflammatory bowel disease (IBD), characterized by infiltration of polymorphonuclear neutrophils (PMNs), increases the risk of colon cancer. PMN activation corresponds to the accumulation of intracellular Lipid Droplets (LDs). As increased LDs are negatively regulated by transcription factor Forkhead Box O3 (FOXO3), we aim to determine the significance of this regulatory network in PMN-mediated IBD and tumorigenesis. Affected tissue of IBD and colon cancer patients, colonic and infiltrated immune cells, have increased LDs’ coat protein, PLIN2. Mouse peritoneal PMNs with stimulated LDs and FOXO3 deficiency have elevated transmigratory activity. Transcriptomic analysis of these FOXO3-deficient PMNs showed differentially expressed genes (DEGs; FDR < 0.05) involved in metabolism, inflammation, and tumorigenesis. Upstream regulators of these DEGs, similar to colonic inflammation and dysplasia in mice, were linked to IBD and human colon cancer. Additionally, a transcriptional signature representing FOXO3-deficient PMNs (PMN-FOXO3_389_) separated transcriptomes of affected tissue in IBD (*p* = 0.00018) and colon cancer (*p* = 0.0037) from control. Increased PMN-FOXO3_389_ presence predicted colon cancer invasion (lymphovascular *p* = 0.015; vascular *p* = 0.046; perineural *p* = 0.03) and poor survival. Validated DEGs from PMN-FOXO3_389_ (*P2RX1, MGLL, MCAM, CDKN1A, RALBP1, CCPG1, PLA2G7*) are involved in metabolism, inflammation, and tumorigenesis (*p* < 0.05). These findings highlight the significance of LDs and FOXO3-mediated PMN functions that promote colonic pathobiology.

## 1. Introduction

Inflammatory bowel disease (IBD), a chronic inflammation of the intestinal tract, includes two distinct clinical entities, Crohn’s disease (CD) and ulcerative colitis (UC) [1]. Both CD and UC are linked to genetic predisposition, impaired barrier function, aberrant microbiota, and dysregulation in the immune system [2,3]. One of the hallmarks of IBD is an excessive infiltration of polymorphonuclear leukocytes (PMNs), also known as neutrophils [4,5]. Moreover, chronic inflammation within IBD may lead to dysplasia, and patients with a history of UC are more predisposed to colon cancer than their healthy counterparts [6,7]. Although PMNs contribute to this inflammation-induced colonic tumorigenesis [8,9], the mechanisms involved in the processes are not fully understood.

PMNs, accounting for ~60% of circulating immune cells, are the first and most rapidly migrating cells to the sites of tissue damage and microbial invasion, where they neutralize microorganisms, recruit other immune cells, and remodel damaged tissues to resolve injury [10,11]. In the intestine, hyperactivation of PMNs results in abnormal immune responses, tissue damage, and aberrant barrier function leading to chronic inflammation [12,13]. Furthermore, in inflamed tissue, PMNs release reactive oxygen and nitrogen species in intestinal epithelial cells, causing DNA mutations that consequently drive tumorigenesis [14,15]. Additionally, augmented PMNs mediate the breakdown of barrier function in the intestine, allowing for the presence of microbiota products [16,17], which can further foster tumorigenic processes in colonic cells. Recent studies have demonstrated that intracellular lipid metabolism is critical in driving PMN activity. Specifically, lipid droplets (LDs), intracellular lipid-storing organelles, are crucial for the development and differentiation of PMNs. Their increase in PMNs is associated with inflammation and bacterial infection [18,19,20,21]. Inflammatory mediators that activate PMNs are also shown to elevate their LDs accumulation [22,23]. Additionally, PMNs in the tumorigenic environment have elevated LDs, which act as an energy source for cancer cell growth and survival, as well as release oxidized lipids that can activate dormant tumor cells and promote metastasis [24,25].

In mouse models of colonic inflammation and inflammatory tumors, LDs are elevated in colonic cells and infiltrated immune cells [26,27,28,29,30]. Our lab demonstrated the existence of a self-regulating negative loop between LDs and transcription factor Forkhead Box O3 (FOXO3) in colonic cells that drive inflammatory and tumorigenic processes [27,29,30,31]. Further, FOXO3 deficiency in the liver leads to hyperlipidemia in mice, with increased hepatic lipid secretion and elevated serum triacylglycerol and cholesterol levels [32]. In addition, FOXO3 cellular function is associated with inflammation and tumorigenesis in the colon [33,34,35,36]. Genome-wide association studies (GWAS) have linked the polymorphism of *FOXO3* (leading to its reduced levels) to the severity of inflammation in IBD [37]. Additionally, FOXO3 can act as a tumor suppressor, and its loss correlates with advanced human colon cancer [33,36]. Given this line of evidence, we hypothesize that this LDs and FOXO3 negative regulatory network in PMNs acts as a promoter of inflammatory tumorigenesis in the colon. These findings provide conceptual advances in understanding mechanisms in PMNs linked to IBD and IBD-facilitated colon cancer.

## 2. Results

### 2.1. Levels of PLIN2 Are Increased in Affected Tissue in IBD and Human Colon Cancer

We have previously demonstrated in the inflamed mouse colon elevated levels of LDs in colonic and infiltrating immune cells [27]. To determine the significance of increased LDs in human intestinal pathobiology, we assessed the levels of PLIN2, the LDs coat protein, in affected tissue obtained from UC and CD patients and in tumor tissue from colon cancer patients. In normal human colonic tissue, immunohistostaining for PLIN2 showed the presence of the protein in the cytosol of colonic epithelial cells along the crypts. In UC and CD, PLIN2 levels were increased by more than 2-fold in the affected tissue, which is significant in both intestinal epithelial cells and infiltrated immune cells (Figure 1A–C). Similar increased PLIN2 levels were found in human colonic tumor tissue (adenocarcinoma, Stage IIB, and Stage IIIC) relative to adjacent normal tissue (Figure 1D,E). These findings demonstrated that LDs are elevated in affected tissue of IBD and human colon cancer as well as in infiltrated immune cells.

### 2.2. Increased Migratory Activity of PMNs Is Associated with Elevated LDs and FOXO3 Deficiency

We have demonstrated the critical role of elevated LDs and loss of FOXO3, an established regulatory network in colonic cells, in driving inflammation and tumorigenesis in the colon, the role of which in immune cells is not well understood. As PMNs are critical players in inflammatory and tumorigenic processes in the colon [5,9,38,39], we investigated the significance of increased LDs and loss of FOXO3 in PMNs’ activity. Intraperitoneal PMNs were obtained from wild type (WT) and FOXO3 knock-out (KO) mice following casein injection [40,41]. After PMNs enrichment, the final fraction contained more than 90% PMNs (FACS assessment of surface marker Ly6G), and peritoneal PMNs in FOXO3 KO mice were increased compared to WT. PMNs were seeded in the upper compartment of a transwell system, and migration activity toward bacterial fMLP or chemokine KC was assessed by quantification of the activity of myeloperoxidase (MPO), an enzyme released by activated PMNs. Oleic acid (OA) stimulation of LDs accumulation in PMNs from WT mice significantly increased their migration, basally stimulated by fMLP and KC (Figure 2A, *p* < 0.05). Further, FOXO3 KO PMNs, relative to WT, had significantly increased migration in response to fMLP, as shown by the MPO activity (Figure 2B, *p* < 0.05). These findings demonstrate that PMNs migration is increased with LDs accumulation and FOXO3 deficiency, suggesting that LDs and FOXO3 regulatory networks in PMNs may play a role in driving inflammation and tumorigenesis.

### 2.3. FOXO3 Deficiency in PMNs Is Associated with Mouse Colonic Inflammation and Dysplasia

Next, we determined in FOXO3-deficient PMNs systemic transcriptional changes and associated molecular pathways driving their activity in the colon. Transcriptional assessment of intraperitoneal PMNs from FOXO3 KO mice showed 212 increased and 242 decreased differentially expressed genes (DEGs) relative to WT (>|0.5|-fold change, FDR < 0.05). The diseases and functions associated with these DEGs include gastrointestinal disorders, metabolic diseases, lipid metabolism, immune responses, immune cell trafficking, inflammatory pathways, and cancer (Figure 3A). To determine the contribution of loss of FOXO3 in PMNs to colonic pathobiology, their upstream regulators of DEGs were identified and compared to transcriptomes from the total colonic tissue of FOXO3-deficient mice, which has exacerbated inflammatory and tumorigenic processes [27,35,42]. We found substantial similarities in upstream regulators of DEGs associated with FOXO3 KO PMNs and total FOXO3 KO colon, which were related to lipid metabolism (eicosapentenoic acid, LEP), immune response (CD40LG, CpG oligonucleotide), inflammatory pathways (SIRT1, HNF-4α), and tumorigenesis (NUPR1, ID1, CREB1, CBFB) (Figure 3B). Next, we determined the significance of FOXO3 KO PMNs in mouse intestinal pathobiology by comparing FOXO3-dependent DEGs in PMNs with publicly available transcriptomes obtained from mouse colons with inflammation and dysplasia (GSE31106). Upstream regulators of FOXO3 KO-dependent DEGs in PMNs were similar to those related to both inflammation and high-grade dysplasia in mouse colon (Figure 3C,D). These shared regulators were linked to metabolism (LDL, FOS, PPAR-γ), immune response (BTK, CD3, BCL6), inflammation (SIRT1, MMP9, TRAF2), and tumorigenesis (MAPK1, CDKN2A, KLF5, Sp1, TGFbeta). These findings revealed the significance of the loss of FOXO3 in PMNs in driving metabolic, inflammatory, and tumorigenic processes in mouse colon.

### 2.4. FOXO3 Deficiency in PMNs Is Associated with Metabolic, Inflammatory, and Tumorigenic Processes in IBD and Human Colon Cancer

Based on our above findings, we sought to determine the significance of the loss of FOXO3 in PMNs in IBD and human colon cancer. The transcriptomes from PMNs of FOXO3 KO mouse have shared regulators with those representing publicly available transcriptomes of affected tissue of different patient cohorts with UC (GSE36807, GSE53306, GSE59071) and CD (GSE59071, GSE95095, GSE102133) (Figure 4A,B). Similar upstream regulators included those involved in metabolism (LEP, NR1H4, PLCG2), immune response (CpG oligonucleotide), inflammatory pathways (C5AR1), and tumorigenesis (NFAT5, RHOA, PTEN, MAPK8). Moreover, DEGs for FOXO3 KO PMNs shared upstream regulators with publicly available transcriptomes from several human colon cancer patient cohorts (GSE141174, GSE8671, GSE9348). These regulators had roles in metabolism (ACOX1, NR1H4), immune response (butyric acid, HIC1), inflammation (ETV5, HNF4-α), and tumorigenesis (Mek, RHOA, EHF, MAPK8, PTEN, LEF1) (Figure 4C). These data demonstrated that loss of FOXO3 in PMNs is important in driving metabolic, inflammatory, and tumorigenic processes in affected tissue of IBD and human colon cancer patients.

### 2.5. Transcriptomic Signature Representing FOXO3-Deficient PMNs Is Highly Prevalent in Transcriptomes of IBD and Human Colon Cancer

To further determine the significance of the FOXO3-deficient PMNs in colonic pathobiology, we established their transcriptional signature, which represents a panel of 389 DEGs (PMN-FOXO_389_), with stringent differential expression and statistical thresholds relative to WT (log2 fold-change >|1.5| and an adjusted *p*-value < 0.001). Unsupervised hierarchical clustering demonstrated that the PMN-FOXO3_389_ signature separated transcriptomes (publicly available data, GSE4183) of affected IBD tissue (UC and CD) and colon cancer samples (adenomas and adenocarcinomas) from the control (unaffected) group (Figure 5A). Additionally, principal component analysis (PCA) showed that in the transcriptome of patients with IBD and colon cancer, the PMN-FOXO3_389_ signature stratified samples of the affected colon from the normal (Figure 5B,C). The PCA index score indicated that PMN-FOXO3_389_-dependent differentiation of the two groups was strongly significant, with *p* = 1.8 × 10^−4^ for IBD (vs. normal) and *p* = 3.7 × 10^−3^ for colon cancer (vs. normal) (Figure 5D,E).

Moreover, we determined the significance of PMN-FOXO3_389_ in a large number of colon cancer patients using publicly available transcriptomes from the TCGA database. In unsupervised hierarchal clustering, PMN-FOXO3_389_ signature separated transcriptomes of colon cancer samples from normal, which was further quantified by PCA index score between two groups (*p* = 2 × 10^−16^) (Figure 6A,B). In addition, PMN-FOXO3_389_ signature separated colon cancer samples according to the probability for invasions in the vasculature (PCA, *p* = 0.046), lymphovasculature (PCA, *p* = 0.015), and perineural space (PCA, *p* = 0.03) (Figure 6C–E). Moreover, Kaplan–Meier estimates showed that the increased presence of PMN-FOXO3_389_ signature in transcriptomes of colon cancer is linked to poor 5-year patient survival rates of 47% (28–65%) when compared to 65% (55–74%) for lower presence (*p* = 8.2 × 10^−4^) (TCGA) (Figure 6F). Together, these data demonstrated that the loss of FOXO3 in PMNs is associated with human colon cancer progression, metastasis, and survival.

### 2.6. FOXO3-Dependent Genes Regulate Metabolic, Inflammatory, and Tumorigenic Processis in PMNs

We identified the top FOXO3 KO-dependent DEGs in PMNs, as shown in the heatmap (Figure 7A) and Table 1. Among these DEGs, we selected for validation genes involved in diverse PMN functions, such as signaling receptors important for their recruitment, adhesion molecules essential for their tissue infiltration and expansion, as well as genes involved in lipid metabolism in PMNs. These DEGs were validated in FOXO3 deficient PMNs (vs. WT) by qPCR (Figure 7B). Specifically, the genes included purinergic receptor P2X 1 (*P2RX1*), which belongs to G-protein-coupled receptors, RalA-binding protein 1 (*RALBP1*), which plays a role in receptor-mediated endocytosis, and melanoma cell adhesion molecule (*MCAM*), as well as cell cycle progression gene 1 (*CCPG1*), which is involved in the immune response to endoplasmic reticulum stress and cyclin-dependent kinase inhibitor 1A (*CDKN1A*). Further, Phospholipase A2 Group VII (*PLA2G7*) is responsible for phospholipid metabolism, and Monoacylglycerol lipase (*MGLL*) catalyzes the conversion of monoacylglycerides to free fatty acids and glycerol. In addition to *MGLL*, adipose triglyceride lipase (*ATGL*) and hormone-sensitive lipase (*HSL*) were lowered in DEGs from FOXO3 KO PMNs (NS). These lipases are responsible for the breakdown of triacylglycerols from LDs [43], suggesting that loss of FOXO3 in PMNs may regulate LDs utilization. These findings establish the importance of FOXO3-mediated gene expression in PMNs that regulate human colonic inflammatory and tumorigenic pathobiology linked to lipid metabolism.

## 3. Discussion

Chronic inflammation in IBD, characterized by massive infiltration of PMNs, is associated with increased risk and progression of colon cancer [4,5,6,7]. Here, we demonstrated that PMNs promote inflammation and inflammatory tumorigenesis in the colon via the LDs and FOXO3 negative regulatory network. Further, we identified the PMN-FOXO3_389_ transcriptional signature, which is increased in IBD and human colon cancer and is highly significant in their pathobiology. Ultimately, we identified FOXO3-dependent differentially expressed genes in PMNs with roles in metabolism, inflammation, and tumorigenesis. Together, these findings establish a novel mechanism in PMNs involving LDs and FOXO3, driving inflammatory and tumorigenesis processes in the colon.

Aberrant PMN function exacerbates inflammation and tumorigenesis in the colon [8,9,14,38]. In DSS-induced colitis in mice, depletion of PMN leads to lowered inflammation and colitis-associated tumorigenesis [9]. In humans, impaired PMN function during chronic inflammation can promote tumorigenesis, as demonstrated in lung and pancreatic cancer [44,45,46]. PMNs can also cluster with cancer cells, aiding tumor growth and metastasis [47,48]. Altered lipid metabolism in PMNs is linked to inflammatory conditions. Accumulation of LDs facilitates early innate response to viral infection through modulation of interferon signaling, in part via TLR7 and TLR9 pathways [23,49]. Similarly, our data showed in FOXO3-deficient PMNs increased *TLR9*, suggesting their sensitivity to viral components. Moreover, this LDs and FOXO3-mediated immune sensitivity to infection is shown in FOXO3 KO colon with increased bacterial LPS sensing *TLR4* [35]. Additionally, LDs accumulation in PMNs is accompanied by elevated inflammatory mediators, which in turn promotes PMN activity [22]. Further, increased LDs in PMNs, due to deficiency of adipose triglyceride lipase (ATGL), also increase PMN activity [19]. For instance, in metastatic breast cancer in the lung, reduced ATGL elicits LDs accumulation in PMNs, promoting the invasive capacity of cancer [24]. Moreover, PMNs can release oxidized lipids that reactivate dormant cancer cells and facilitate tumor recurrence [25]. Hence, PMNs with elevated LDs can facilitate inflammatory and tumorigenic processes in various tissue, including the colon.

Moreover, we demonstrated that in PMNs, a negative regulatory network of LDs and FOXO3 might be one of the mechanisms of colonic inflammation and tumorigenesis within IBD and colon cancer. We have previously demonstrated that this regulatory network drives pathobiological processes in human colonic cells [27,29,30,31]. In mice, global FOXO3 deficiency increases PMNs in the spleen, bone marrow, blood, and colon [35,50]. Similarly, FOXO3 inactivation results in elevated PMNs and aberrant immune response in bronchial epithelia [51]. We speculate that the increased infiltration and elevated transmigration of these PMNs are supported by their FOXO3-dependent metabolic reprogramming. Further, our findings revealed, in FOXO3 deficient PMNs, decreased farnesoid X receptor (FXR; *NR1H4*), a sensor of intracellular bile acid levels. FXR protects against bile acid toxicity, and it is reduced in CD patients [52].

Further, we identified several FOXO3-dependent differentially expressed genes in PMNs involved in metabolism, inflammation, and tumorigenesis. Specifically, *MGLL* is involved in fatty acid metabolism and plays roles in tumorigenesis and metastasis [53]. *P2RX1* is linked to the modulation of microbiota and the alleviation of inflammation in colitis [54]. *PLA2G7*, which is shown to be reduced in FOXO3 KO PMNs, is associated with inflammation [55]. Two other DEGs, *MCAM* and *CDKN1A*, in FOXO3 KO PMNs, are involved in tumorigenesis [56,57]. *RALBP1* has a key effector function in cancer cell survival [58]. Moreover, we found several DEGs in FOXO3 KO PMNS involved in lipid metabolisms linked to LDs utilization. *MGLL*, *ATGL*, and *HSL* lipase, critical for the breakdown of triacylglycerols stored in LDs [43], are decreased in FOXO3 KO PMNs. ATGL deficiency in PMNs leads to increased LDs and hyperactivation [19]. These findings suggest that although FOXO3 deficiency in colonic epithelial cells leads to increased LDs biosynthesis [27,30,31,35], in PMNs, it may lead to increased LDs by lowering their utilization. In addition, the significance of the loss of FOXO3 in driving PMN function is at least multifactorial, and while multiple differentially expressed genes were insignificantly altered, their synergistic effects on the transcriptomic and metabolic remodeling of PMN drive colonic inflammation and tumorigenesis.

One of the hallmarks of IBD is excessive infiltration of PMNs that intensify the inflammatory pathobiology, leading to tumorigenesis [4,5,12,13,15], the mechanisms of which are poorly understood. Here, we showed that PMNs drive colonic inflammation and tumorigenesis by facilitating a self-reinforced LDs and FOXO3 negative regulatory network. Additionally, we demonstrated the significance of this mechanism in both IBD and colon cancer. It is important to highlight that, in addition to LDs and FOXO3 in PMNs, complex processes drive IBD and colonic tumorigenesis that include multiple cells, different pathways, and regulators. Further, this network altered the expression of multiple genes, which, even if insignificant, may orchestrate systemic changes in PMNs functions. As PMNs are highly sensitive cells to multiple stimuli and may initially be protective to infiltrated tissue, it is required to further delineate these mechanisms in context to other cells and regulators. Together, our findings establish an important mechanism that drives PMN activity, bringing us one step closer to solving the complex puzzle of IBD and inflammatory colon cancer.

## 4. Materials and Methods

### 4.1. Human IBD and Colon Cancer Samples

Tissue microarray samples included human intestinal tissue representing normal, Inflammatory Bowel Diseases (Ulcerative Colitis and Crohn’s Disease), and colon adenocarcinoma with tumor-adjacent colonic mucosa (tissuearray.com LLC). These tissues were obtained from different individuals, males and females, aged 20 to 66 (total *n* = 10, *n* = 2 of each for normal, UC, CD, tumor-adjacent mucosa, and adenocarcinoma).

Publicly available transcriptomes were obtained for control and affected tissue from three different patient cohorts of UC (total *n* = 132; GSE36807, GSE53306, GSE59071) and CD (total *n* = 91; GSE59071, GSE95095, GSE102133). Publicly available transcriptomic data from tumor tissue obtained from three colon cancer patient cohorts (total *n* = 162; GSE141174, GSE8671, GSE9348) were used. Moreover, publicly available transcriptomic data of colon cancer patients, including normal controls, were also utilized (*n* = 498, TCGA). These data were acquired by using NCBI’s GEO2R.

### 4.2. Mice

Mice, strain C57BL/6, male and female, were housed under pathogen-free conditions at Tulane University School of Medicine. Both wild type (WT) and FOXO3 knock-out (FOXO3 KO) mice had free access to a standard chow diet and water. All littermates were genotyped to identify homozygous WT and FOXO3 KO, according to the guidelines of Tulane Institutional Animal Care and Use Committee [34]. Transcriptomic data from the colon of FOXO3 KO and WT mice were acquired as described before (*n* = 5–6 mice per group) [35].

Moreover, publicly available transcriptomes obtained from mice with colonic inflammation (*n* = 3) and dysplasia (*n* = 3) were utilized (GSE31106) [59].

### 4.3. Histological Analysis

Immunohistostaining against antibody PLIN2 (LS Bio, Seattle, WA, USA) of tissue microarray with samples from IBD and human colon cancer patients was performed by The Pathology Core Laboratory at Tulane University Health Sciences Center as described previously [35]. Images were obtained using the Phenolmager fusion slide scanner (Akoya, Menlo Park, CA, USA) and Phenochart 1.2.0 software. Images were quantified utilizing ImageJ/Fiji 2.1.0 by performing spectrum deconvolution for the separation of DAB (diaminobenzidine) color spectra. The DAB image was then analyzed pixel by pixel for immunohistochemistry quantification.

### 4.4. Mouse Peritoneal Polymorphonuclear Neutrophils (PMNs)

Experimental mice (six to eight weeks old) were injected intraperitoneally with 1 mL of sterile casein (Sigma, St. Louis, MO, USA) solution, followed by a second injection the next day, which caused a response of peritoneal PMNs [41]. Three hours after the second injection, mouse peritoneal cells were harvested from the abdominal cavity and pelleted (200× *g* for 3 times). Next, PMNs were isolated from the peritoneal fluid using histopaque separation media using a density gradient centrifugation method [40].

### 4.5. Transmigration Assay

Mice peritoneal PMNs, 10 × 10^6^ cells in assay buffer, HEPES containing 10 mM glucose, 0.1% BSA, pH 7.4, and 1% penicillin/streptomycin, was incubated with oleic acid (OA) (50 µM), for 2 h, and were placed on the top of transwells, an 8 mM pore size polycarbonate filter. In the lower compartments of transwells, 1 mL of the assay media with N-Formylmethionyl-leucyl-phenylalanine (fMLP) (1 mM) or chemokine KC (29 ng/mL) was added. After 30 min, at 37 °C, assay media from the lower compartment was collected and centrifuged at 400× *g* for 5 min to assess for migrated PMNs.

### 4.6. Myeloperoxidase (MPO) Colorimetric Activity Assay

Enzymatic activity of the activated PMN’s marker, known as myeloperoxidase (MPO), was quantified in PMNs from WT (*n* = 6) and FOXO3 KO (*n* = 6) mice (*n* = 3 wells per mouse) using MPO Colorimetric Activity Assay Kit according to the manufacturer’s protocol (Sigma, St. Louis, MO, USA). Briefly, collected PMN samples were rapidly homogenized in MPO assay buffer and centrifuged at 13,000× *g* for 10 min at 4 °C to remove insoluble material. These samples were then plated in a 96-well plate, further assessed, and used for colorimetric detection of MPO activity at 412 nm.

### 4.7. RNA Isolation and cDNA Synthesis

Total RNA from harvested PMNs (*n* = 5–6 mice per group) was isolated using the miRNeasy kit (Zymo Research, Irvine, CA, USA), following the manufacturer’s instructions. First, RNA was evaluated for quality using Agilent Bioanalyzer (Agilent Technologies, Santa Clara, CA, USA). Samples having RNA integrity numbers (RIN) of more than 8 were utilized. RNA was then reverse transcribed to cDNA with qScript cDNA SuperMix (Quantabio, Beverly, MA, USA) and used for qPCR as described previously [35]. The primers used for the amplification of mouse cDNA are as follows: (mP2RX1-FOR 5′-GACAAACCGTCGTCACCTCT-3′, mP2RX1-REV 5′-TCACGTTCACCCTCCCCA-3′, mMGLL-FOR 5′-TTTCCTTCCCTAAGCGGTCG-3′, mMGLL-REV 5′-GGGGTCTTTAGGCCCTGTTT-3′, mMCAM-FOR 5′-CGGGTGTGCCAGGAGAG-3′, mMCAM-REV 5′-GGTTCCTCTGGGGCTTTGAA-3′, mCDKN1A-FOR 5′-GCAGAATAAAAGGTGCCACAGG-3′, mCDKN1A-REV 5′-AGAGTGCAAGACAGCGACAA-3′, mRALBP1-FOR 5′-CTCGTCCTGTTCTGTCCCAA-3′, mRALBP1-REV 5′-ACCTATCCATTACACCAGTGCC-3′, mCCPG1-FOR 5′-AGAAAGCAGCGCAAACAACA-3′, mCCPG1-REV 5′-CTAGGCTGAGATGAAAAGACGGG-3′, mPLA2G7-FOR 5′-TCCCTGGAGCTAGTGTTGTG-3′, mPLA2G7-REV 5′-TGGCTTCAGTTTGATGTTCTGGT-3′. The comparative Ct method was used to determine mRNA expression with actin as a housekeeping control. cDNA was quantified using the C1000 Thermal Cycler system (Bio-Rad, Hercules, CA, USA) and PerfeCTa SYBR Green FastMix (Quantabio, Beverly, MA, USA).

### 4.8. RNA Sequencing and Differential Expression Testing

RNA sequencing (RNAseq) was performed as described previously [35]. Sequencing data are submitted in NCBI’s Sequence Read Archive and are available under GSE234072 study accession number. Transcriptomic analysis for RNAseq was performed using Ingenuity Pathway Analysis (IPA) (Qiagen, Germantown, MD, USA). Differentially expressed genes (DEGs) with an expression threshold of >|0.5|-fold change relative to control and a false discovery rate (FDR) of less than 0.05 were evaluated in IPA. Clustered heatmaps of z-scaled transcripts per million (TPM) values for the top DEGs across all samples were obtained using a Python data visualization package (Seaborn v0.12.0). 

### 4.9. Hierarchical Clustering

Hierarchical clustering of transcriptomes among experimental groups was performed using an uncentered correlation as a symmetric matrix, as described before [28].

### 4.10. Principal Component Analysis

Principal component analysis (PCA) of FOXO3-deficient PMNs’ signatures with IBD (GSE4183) and human colon cancer (TCGA) transcriptomes was performed with the FactoMineR R package with the PCA function, as described before [28].

### 4.11. Statistical Analysis

All results are represented as mean ± SD. The statistical analysis of experiments was carried out by Student’s unpaired *t*-test or through ANOVA for one-way analysis of variance in Graph Pad Software. A *p*-value of <0.05 was considered significant.

## Figures and Tables

**Figure 1 ijms-24-09730-f001:**
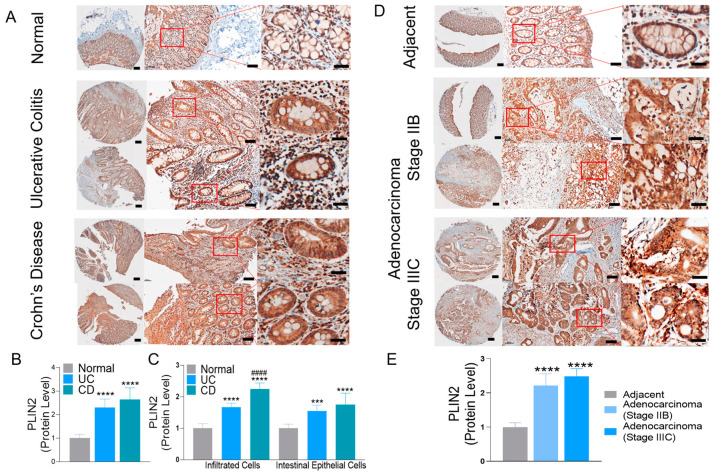
LDs’ coat protein PLIN2 levels are increased in affected human colonic tissue obtained from (**A**) Ulcerative Colitis (UC), Crohn’s Disease (CD), and (**D**) Colon Cancer patients. PLIN2 levels in tissue were determined by immunohistostaining (IHC) (tissue array). Areas selected in red boxes are further magnified sections (scale bar 800 µM, 100 µM, 20 µM) (**B**,**C**,**E**) Graphs represent the quantification of IHC of PLIN2 using ImageJ/Fiji 2.1.0, performing spectrum deconvolution for separation of DAB color spectra. Total number of patients (*n* = 10) included normal (*n* = 2), UC (*n* = 2), CD (*n* = 2), tumor-adjacent mucosa (*n* = 2), and adenocarcinoma (*n* = 2). Quantification included 5–8 separate areas/samples (**** *p* < 0.0001, *** *p* < 0.001 vs. normal control, #### *p* < 0.0001 vs. UC samples).

**Figure 2 ijms-24-09730-f002:**
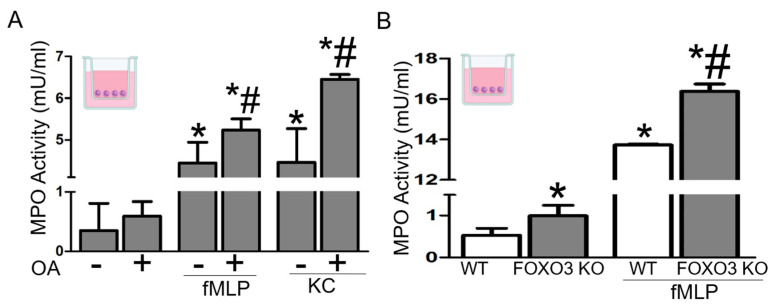
Increased LDs and FOXO3 deficiency in PMNs facilitate their migratory activity. (**A**) MPO activity showing transmigration of mouse peritoneal PMNs with Oleic acid (OA) stimulated LDs. fMLP and KC are used as PMNs chemoattractants. (*n* = 6 mice per group, *n* = 3 wells per mouse, * *p* < 0.05 vs. control, # *p* < 0.05 vs. fMLP or KC without OA) (**B**) MPO activity showing transmigration of peritoneal PMNs from FOXO3 KO mouse relative to wild type (WT) in response to fMLP. *(n* = 6 mice per group, *n* = 3 wells per mouse, * *p* < 0.05 vs. WT, # *p* < 0.05 vs. WT treated with fMLP).

**Figure 3 ijms-24-09730-f003:**
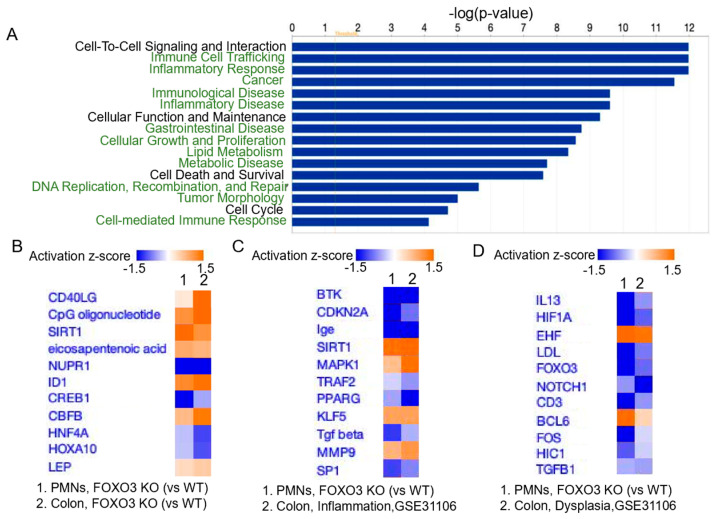
FOXO3 deficiency in PMNs is linked to mouse colonic pathobiology. (**A**) Top diseases and pathways associated with FOXO3 KO in PMNs relative to control (*p* < 0.05, IPA). (**B**) Top upstream regulators of DEGs mediated by FOXO3 deficiency in PMNs, shared with FOXO3 KO mouse colon (IPA, *n* = 3–5 mice per group, *p* < 0.05). (**C**,**D**) Top regulators of FOXO3 KO-dependent DEGs in PMNs shared with regulators in mouse colon with inflammation and high-grade dysplasia (*n* = 3 mice per group, IPA, *p* < 0.05, IPA, GSE31106).

**Figure 4 ijms-24-09730-f004:**
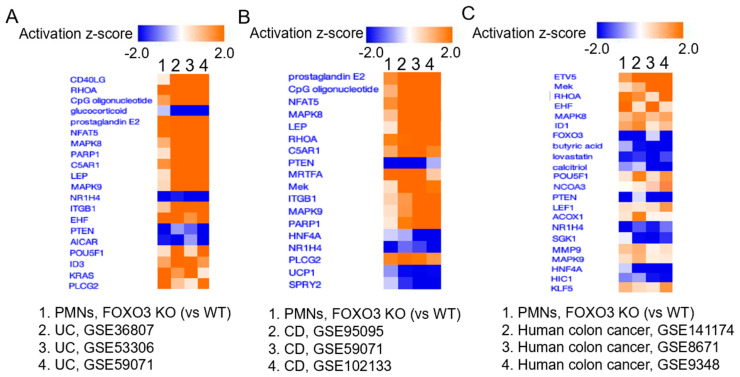
FOXO3 deficiency in PMNs is linked to drivers of IBD and human colon cancer. Top regulators of DEGs in FOXO3 KO PMNs similar to regulators in (**A**) UC, (**B**) CD, and (**C**) human colon cancer. Publicly available data from three different patient cohorts for each, UC (GSE36807, GSE53306, GSE59071, total *n* = 132), CD (GSE59071, GSE95095, GSE102133, total *n* = 91), and human colon cancer (GSE141174, GSE8671, GSE9348; total *n* = 162) were utilized (IPA, *p* < 0.05).

**Figure 5 ijms-24-09730-f005:**
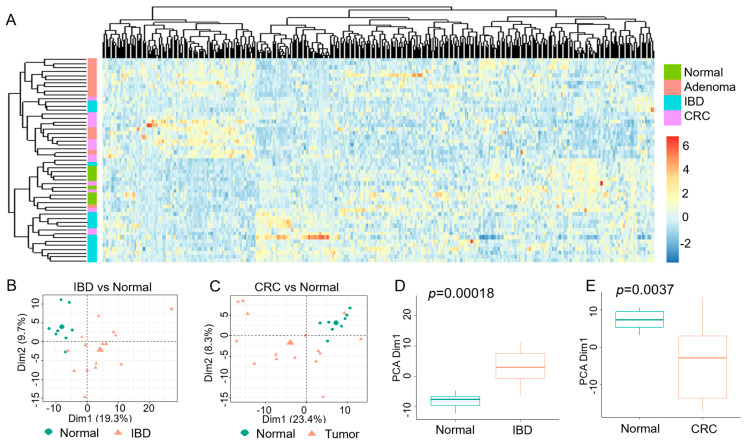
Transcriptional signature for FOXO3-dependent PMNs in IBD and human colon cancer. (**A**) Hierarchical clustering, as shown by a heatmap, revealing the separation of transcriptomes representing IBD (blue) and colon cancer (adenomas (pink) and adenocarcinomas (CRC, violet)) from normal (green) by the PMNs-FOXO3_389_ signature. PMN-FOXO3_389_ signature on *x*-axis and human samples on *y*-axis. (**B**,**C**) FOXO3_389_ signature was used to perform principal component analysis (PCA) of active IBD and matched control transcriptomes, as well as human colon cancer and control transcriptomes, to estimate variation between samples. (**D**,**E**) Two-axis values of the PCA showed that PMN-FOXO3_389_ significantly differentiated IBD from matched control tissue (*p* = 0.00018) and colon cancer tissue from control (*p* = 0.0037) (*n* = 53, GSE4183).

**Figure 6 ijms-24-09730-f006:**
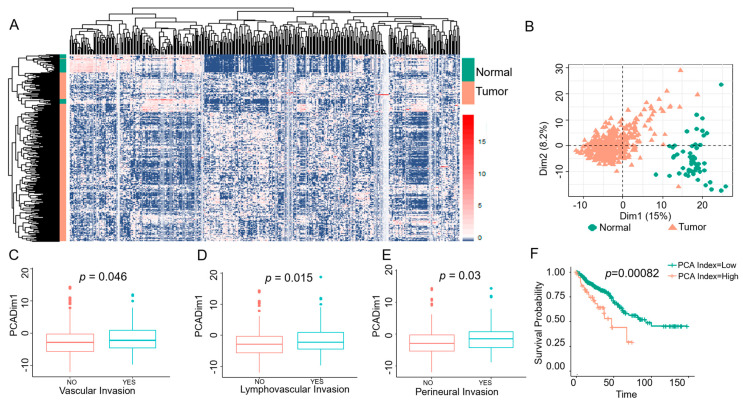
Transcriptional signature for FOXO3-dependent PMNs in human colon cancer. (**A**,**B**) Hierarchical clustering, shown by heatmap, revealed distinct clusters separated by the PMN-FOXO3_389_ signature differentiating human colon cancer (pink) from matched normal (green) transcriptomes (*n* = 432, TCGA). PMN-FOXO3_389_ signature on *x*-axis and human samples on *y*-axis. PMN-FOXO3_389_ stratify the colon tumor samples from normal. (**C**–**E**) Increased PMN-FOXO3_389_ signature presence in colon cancer patients is associated with metastasis (lymphovascular invasion, vascular invasion, perineural invasion. (**F**) Increased PMN-FOXO3_389_ signature presence in transcriptomes of colon cancer patients revealed worse overall 5-year survival with a high PCA index score (pink)) compared to low score (green) (*p* = 0.00082).

**Figure 7 ijms-24-09730-f007:**
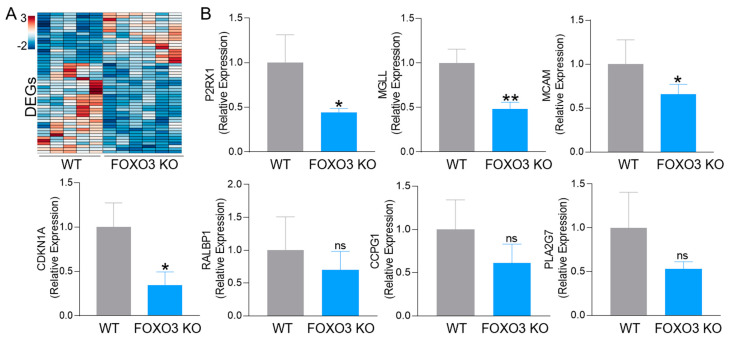
FOXO3-mediated differentially expressed genes in PMNs. (**A**) A heatmap of the top DEGs specific to FOXO3-deficient PMNs relative to control (>|0.5|-fold change, FDR < 0.05). (**B**) Validation of select FOXO3 dependent *P2RX1*, *MGLL*, *MCAM*, *CDKN1A*, *RALBP1*, *CCPG1* and *PLA2G7* transcripts in peritoneal PMNs (qPCR, *n* = 3 mice per group, * *p* < 0.05, ** *p* < 0.01, ns = not significant).

**Table 1 ijms-24-09730-t001:** Differentially expressed genes in FOXO3-deficient mouse PMNs relative to control (*n* = 5–6, FC ≥ |1.5|, *p* < 0.5).

	Gene	Gene Name	FC	*p*-Value
1	*UBE2D2A*	Ubiquitin-conjugating enzyme E2D 2A	4.98	8.9 × 10^−3^
2	*CATSPERE2*	Cation channel sperm associated auxiliary subunit epsilon 2	4.77	1.4 × 10^−2^
3	*LALBA*	Lactalbumin alpha	3.58	4.8 × 10^−2^
4	*DLL3*	Delta-like canonical Notch ligand 3	3.31	1.2 × 10^−2^
5	*ANO2*	Anoctamin 2	3.30	4.2 × 10^−2^
6	*ASCL4*	Achaete-scute family bHLH transcription factor 4	3.24	4.9 × 10^−2^
7	*HIST1H3B*	Histone cluster 1, H3b	3.17	2.4 × 10^−2^
8	*CCDC85A*	Coiled-coil domain containing 85A	3.12	1.7 × 10^−2^
9	*FAM198A*	Family with sequence similarity 198 member A	3.05	1.9 × 10^−2^
10	*CDKL1*	Cyclin-dependent kinase-like 1	3.00	2.0 × 10^−2^
11	*GBP2B*	Guanylate binding protein 2b	2.56	3.0 × 10^−2^
12	*MYRFL*	Myelin regulatory factor-like	2.47	5.3 × 10^−2^
13	*MRAP2*	Melanocortin 2 receptor accessory protein 2	2.43	1.6 × 10^−2^
14	*SYNPO2*	Synaptopodin 2	2.37	4.4 × 10^−3^
15	*RNF183*	Ring finger protein 183	2.37	1.4 × 10^−2^
16	*CACNA1H*	Calcium voltage-gated channel subunit alpha1 H	2.35	2.6 × 10^−2^
17	*DLG2*	Discs large MAGUK scaffold protein 2	2.26	4.9 × 10^−2^
18	*LRIG3*	Leucine-rich repeats and immunoglobulin-like domains 3	2.25	5.7 × 10^−3^
19	*MAP3K19*	Mitogen-activated protein kinase kinase kinase 19	2.22	1.7 × 10^−2^
20	*TECTA*	Tectorin alpha	2.14	2.1 × 10^−2^
21	*CUBN*	Cubilin	1.98	1.0 × 10^−2^
22	*LDHD*	Lactate dehydrogenase D	1.92	1.8 × 10^−2^
23	*IQCN*	IQ motif containing N	1.87	4.0 × 10^−2^
24	*AMD2*	S-adenosylmethionine decarboxylase 2	1.82	7.8 × 10^−4^
25	*SNORD15A*	Small nucleolar RNA, C/D box 15A	1.78	2.7 × 10^−2^
26	*ASTN2*	Astrotactin 2	1.67	7.0 × 10^−3^
27	*MIR146*	MicroRNA 146	1.65	3.8 × 10^−4^
28	*GBP11*	Guanylate binding protein 11	1.65	4.3 × 10^−2^
29	*GUCY2C*	Guanylate cyclase 2C	1.64	6.0 × 10^−6^
30	*MDRL*	Mitochondrial dynamic related lncRNA	1.59	4.8 × 10^−2^
31	*ADAMTS13*	ADAM metallopeptidase with thrombospondin type 1 motif 13	1.56	4.1 × 10^−2^
32	*RAPGEF4*	Rap guanine nucleotide exchange factor 4	1.56	3.4 × 10^−2^
33	*STX1A*	Syntaxin 1A	1.55	1.0 × 10^−2^
34	*NECTIN3*	Nectin cell adhesion molecule 3	1.54	3.9 × 10^−2^
35	*SLC22A1*	Solute carrier family 22 member 1	1.54	7.1 × 10^−3^
36	*MEX3A*	Mex−3 RNA binding family member A	1.52	5.0 × 10^−2^
37	*RALBP1*	RalA-binding protein 1	−0.44	9.0 × 10^−5^
38	*CDKN1A*	Cyclin-dependent kinase inhibitor 1A	−0.55	1.0 × 10^−3^
39	*PLA2G7*	Phospholipase A2 Group VII	−0.6	1.0 × 10^−3^
40	*TLR9*	Toll Like Receptor 9	−0.8	1.7 × 10^−2^
41	*CCPG1*	Cell cycle progression gene 1	−1.12	2.0 × 10^−3^
42	*MGLL*	Monoacylglycerol lipase	−1.29	3.0 × 10^−3^
43	*GFI1B*	Growth factor independent 1B transcriptional repressor	−1.50	2.1 × 10^−3^
44	*HGF*	Hepatocyte growth factor	−1.52	6.1 × 10^−3^
45	*BATF2*	Basic leucine zipper ATF-like transcription factor 2	−1.52	2.2 × 10^−2^
46	*P2RX1*	Purinergic receptor P2X 1	−1.53	1.4 × 10^−3^
47	*ZFP469*	Zinc finger protein 469	−1.56	1.0 × 10^−2^
48	*TMEM26*	Transmembrane protein 26	−1.56	7.0 × 10^−3^
49	*HUNK*	Hormonally up-regulated Neu-associated kinase	−1.58	2.2 × 10^−2^
50	*SIGLECF*	Sialic acid binding Ig-like lectin F	−1.58	7.4 × 10^−3^
51	*FAIM2*	Fas apoptotic inhibitory molecule 2	−1.59	2.0 × 10^−2^
52	*MCAM*	Melanoma cell adhesion molecule	−1.61	1.9 × 10^−3^
53	*MYLK3*	Myosin light chain kinase 3	−1.62	1.2 × 10^−2^
54	*IL4*	Interleukin 4	−1.65	5.0 × 10^−3^
55	*CCR3*	C-C motif chemokine receptor 3	−1.67	8.4 × 10^−3^
56	*POM121L2*	POM121 transmembrane nucleoporin like 2	−1.85	1.1 × 10^−2^
57	*PLA2G3*	Phospholipase A2 group III	−2.12	7.6 × 10^−3^
58	*SLC27A2*	Solute carrier family 27 member 2	−2.35	3.6 × 10^−3^
59	*IL13*	Interleukin 13	−2.54	5.4 × 10^−3^
60	*PDK4*	Pyruvate dehydrogenase kinase 4	−2.54	1.1 × 10^−2^
61	*CTSG*	Cathepsin G	−2.80	3.3 × 10^−3^
62	*TNXB*	Tenascin XB	−2.85	3.6 × 10^−3^
63	*VSNL1*	Visinin like 1	−3.16	1.7 × 10^−2^
64	*MPO*	Myeloperoxidase	−3.37	2.0 × 10^−3^
65	*TDG-PS*	Thymine DNA glycosylase, pseudogene	−3.38	5.6 × 10^−4^
66	*BPI*	Bactericidal permeability-increasing protein	−3.40	2.2 × 10^−2^
67	*ELANE*	Elastase, neutrophil expressed	−3.54	5.2 × 10^−3^
68	*CAPN1*	Calpain 1	−24.1	1.3 × 10^−15^

## Data Availability

The data presented in this study are available on request from the corresponding author. RNA sequencing of experimental PMNs will be submitted to NCBI’s Archive to be publicly available.

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
