# Peer review of "FOXO3 Deficiency in Neutrophils Drives Colonic Inflammation and Tumorigenesis"

_ijms, 2023, doi:10.3390/ijms24119730_

Round 1
Reviewer 1 Report
The manuscript from Ghimire et al. investigates the effect of FOXO3 loss in neutrophil populations. Using transcriptional analysis of peritoneal neutrophils, the authors identify differentially expressed genes which may affect the development of colonic disease. The authors utilize a number of human disease samples, through both tissue microarrays as well as publicly available datasets, to show that FOXO3 deficiency drives genetic alterations which may play a role in colonic inflammation and tumorigenesis. However, some concerns dampen overall enthusiasm, which are detailed below.
- -In Figure 1, the authors note PLIN2 cytoplasmic staining in epithelial cells, yet focus solely on the neutrophil biology in the later studies. It would be interesting to know whether PLIN2 staining is increased specifically in the infiltrating immune cells or if this is only seen in epithelial cells.
- -In the first two figures, the authors note changes in lipid droplet formation and neutrophil migration induced by FOXO3 loss. Did the transcriptomic analysis identify any changes in FOXO3 deficient neutrophils that may contribute to these changes in biology? Was PLIN2 expression altered in the transcriptomic analysis?
- Was the FOXO3-389 signature associated with higher or lower survival in Figure 6? Based on the color scheme, to this reviewer it appears the signature is associated with various types of invasion, but with higher overall survival. If so, can the authors speculate why this may be the case? If not, a legend or better overall explanation of the color schemes may be helpful, particularly as the colors are used for both tumor/normal distinctions as well as low/high FOXO3 signature within the same figures.
- -It is often unclear how many biological and technical replicates were used throughout the manuscript. For example, how many samples were analyzed per group for the PLIN2 IHC presented in figure 1? N is listed as 3 for the qPCR presented in Figure 7, was this 3 biological or technical replicates? Were technical replicates done for the migration studies?
Author Response
Please see attached response.

Reviewer 2 Report
Dear Authors,
congratulations on your work. It is an interesting topic.
My question is: what are limitations of your study? It is an important thing to see in the article. A paragraph is needed.
Minor spelling check
Author Response
Please see attached response.

Reviewer 3 Report
Review or the manuscript by Jenisha Ghimire et al.
Issues detected:
1. Add introductory information on FOXO3 involvement in lipid and CHO metabolism.
2. Figure5 plots B-E, for sake of consistency, use the same colour for control and malignant (IBD, tumour) samples
3. Line 187, concerns Fig 5, plots B & C: add to the caption description of the visualization (e.g. “PCA plots of…”
4. Table 1, item 53 is FoxO3. Why it shows only 2.5-fold down regulation in your FOXO3-KO cells? Were your mice rather Knock-downs than KOs? Have you trully verified the expression of FoxO3 before the experiment with PMNs?
5. Chapter 2.6, Why those genes (P2RX1, MGLL, MCAM, CDKN1A, RALBP1, CCPG1 and PLA2G7) were selected for the qPCR? There is not rationale given. The decrease in relative expression as measured by is rather small or not significant (Fig 7). Why there are no upregulated genes in the verification pool? MGL is an enzyme which works at the end of lipolysis of LDs. The process is mainly regulated by the proceeding lipases such as ATGL, which would be a one of the first targets to check by PCR and/or to report among the DEGs. Especially, in case of you using ATGL to explain some of your data and conclusions. As it is not the case, I recommend additional measurements
6. Line 328 and other, use μ symbol for micro instead of “u”
7. Line 250, the sentence starting with “Specifically, increased LDs accumulation in PMNs…” suggests causality between LD and antimicrobial activity. The quoted paper by Monson et al. suggest reverse relationship, but it wasn’t proven to act the other way around. What is the expression of TLRs in PMN’s from your KO-mice?
8. Line 272, the claim “another FOXO3-dependent gene in PMNs, PLA2G7, is associated with inflammation.” Is not supported by your data. The qPCR difference in expression for PLA2G7 is not significant and it isn’t on your top DEG list.
9. Line 274, The same as above The expression data from RNAseq isn’t provided and qPCR shows (Fig 7) non-significant results for RALBP1 and CCPG1.
10. An interesting result wasn’t approached and discussed at all. The downregulation of NR1H4. This TF was only named once, although it was shown in Fig 5 to be one of characteristic regulators. The NR1H4 is also known as FXR – farnesoid receptor X, which regulates the bile acids metabolism. The relationship between FXR, bile acids and IBD was already observed and has been discussed several times in recent years. Worth to include in your discussion.
Overall, it is interesting paper, which should be published, but requres extensive modifications.
Author Response
Please see attached response.

Round 2
Reviewer 1 Report
The authors have sufficiently addressed my concerns.
Reviewer 3 Report
2nd review:
The changes and answer are satisfactory. The manuscript can be published in it's current form.
A comment on nomenclature:
Your paper is focused on differences in abundance of RNA. It worth to remember that elevated abundance, possibly through increased gene expression, does not translates 1:1 into elevated copy number of encoded protein. In your manuscript there several sentences like "...our data showed in FOXO3-deficient PMNs increased TLR9,...". It can be helpful to explicitly state whether you are referring to a gene (italicized caps) or protein (caps), particularly within sentences in which both a gene and its product are mentioned (e.g. “We quantified TLR9 gene expression and TLR9 protein levels ...“). Usually, RNA follows the DNA formatting. Also, you could selectively use the term “expression” when referring to genes and the term “levels” when referring to RNA or proteins.